# Rotated Binary Neural Network

**Mingbao Lin** [1]    **Rongrong Ji** [1,2,3*]    **Zihan Xu** [1]    **Baochang Zhang** [4]
**Yan Wang** [5]    **Yongjian Wu** [6]    **Feiyue Huang** [6]    **Chia-Wen Lin** [7]
[1] Media Analytics and Computing Lab, Department of Artificial Intelligence,
School of Informatics, Xiamen University
[2] Institute of Artificial Intelligence, Xiamen University
[3] Peng Cheng Lab        [4] Beihang University        [5] Pinterest
[6] Tencent Youtu Lab        [7] National Tsing Hua University

## Abstract

Binary Neural Network (BNN) shows its predominance in reducing the complexity of deep neural networks. However, it suffers severe performance degradation. One of the major impediments is the large quantization error between the full-precision weight vector and its binary vector. Previous works focus on compensating for the norm gap while leaving the angular bias hardly touched. In this paper, for the first time, we explore the influence of angular bias on the quantization error and then introduce a Rotated Binary Neural Network (RBNN), which considers the angle alignment between the full-precision weight vector and its binarized version. At the beginning of each training epoch, we propose to rotate the full-precision weight vector to its binary vector to reduce the angular bias. To avoid the high complexity of learning a large rotation matrix, we further introduce a bi-rotation formulation that learns two smaller rotation matrices. In the training stage, we devise an adjustable rotated weight vector for binarization to escape the potential local optimum. Our rotation leads to around 50% weight flips which maximize the information gain. Finally, we propose a training-aware approximation of the sign function for the gradient backward. Experiments on CIFAR-10 and ImageNet demonstrate the superiorities of RBNN over many state-of-the-arts. Our source code, experimental settings, training logs and binary models are available at https://github.com/lmbxmu/RBNN.

## 1   Introduction

The community has witnessed the remarkable performance improvements of deep neural networks (DNNs) in computer vision tasks, such as image classification [26, 19], object detection [39, 20] and semantic segmentation [35, 33]. However, the cost of massive parameters and computational complexity makes DNNs hard to be deployed on resource-constrained and low-power devices.

To solve this problem, many compression techniques have been proposed including network pruning [30, 14, 29], low-rank decomposition [12, 43, 18], efficient architecture design [24, 42, 7] and network quantization [28, 2, 21], *etc*. In particular, network quantization resorts to converting the weights and activations of a full-precision network to low-bit representations. In the extreme case, a binary neural network (BNN) restricts its weights and activations to only two possible values ($-1$ and $+1$) such that: 1) the network size is $32\times$ less than its full-precision counterpart; 2) the multiply-accumulation convolution can be replaced with the efficient xnor and bitcount logics.

Though BNN has attracted great interest, it remains a challenge to close the accuracy gap between a full-precision network and its binarized version [38, 6]. One of the major obstacles comes at the large

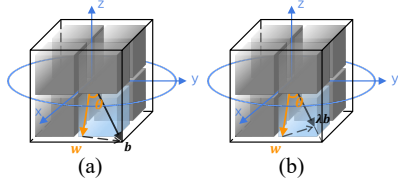

(a)          (b)

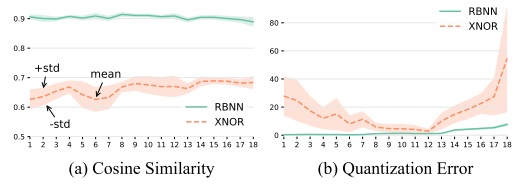

(a) Cosine Similarity          (b) Quantization Error

Figure 1: (a) Early works [8, 9] suffer from large quantization error caused by both the norm gap and angular bias between the full-precision weights and its binarized version. (b) Recent works [38, 37] introduce a scaling factor to reduce the norm gap but cannot reduce the angular bias, *i.e.*, $\theta$. Therefore the quantization error $\|\mathbf{w}\sin\theta\|^2$ is still large when $\theta$ is large.

Figure 2: Cosine similarity and quantization error in various layers of ResNet-20. (a) Our RBNN achieves a significantly higher cosine similarity between the full-precision weight and its binarization than XNOR-Net [38] does, implying fewer angular bias. (b) XNOR-Net suffers great quantization error while RBNN leads to a much smaller one.

quantization error between the full-precision weight vector $\mathbf{w}$ and its binary vector $\mathbf{b}$ [8, 9] as shown in Fig. 1(a). To solve this, state-of-the-art approaches [38, 37] try to lessen the quantization error by introducing a per-channel learnable/optimizable scaling factor $\lambda$ to minimize the quantization error:

$$\min_{\lambda, \mathbf{b}} \|\lambda\mathbf{b} - \mathbf{w}\|^2. \tag{1}$$

However, the introduction of $\lambda$ only partly mitigates the quantization error by compensating for the norm gap between the full-precision weight and its binarized version, but cannot reduce the quantization error due to an angular bias as shown in Fig. 1(b). Apparently, with a fixed angular bias $\theta$, when $\lambda\mathbf{b} - \mathbf{w}$ is orthogonal to $\lambda\mathbf{b}$, Eq. (1) reaches the minimum and we have

$$\|\mathbf{w}\sin\theta\|^2 \leq \|\lambda\mathbf{b} - \mathbf{w}\|^2, \tag{2}$$

Thus, $\|\mathbf{w}\sin\theta\|^2$ serves as the lower bound of the quantization error and cannot be diminished as long as the angular bias exists. This lower bound could be huge with a large angular bias $\theta$. Though the training process updates the weights and may close the angular bias, we experimentally observe the possibility of this case is small, as shown in Fig. 2. Thus, it is desirable to reduce this angular error for the sake of further reducing the quantization error. Moreover, the information of BNN learning is upper-bounded by $2^n$ where $n$ is the total number of weight elements and the base 2 denotes the two possible values in BNN [32, 37]. Weight flips refer to that positive value turns to $-1$ and vice versa. It is easy to see that when the probability of flip achieves 50%, the information reaches the maximum of $2^n$. However, the scaling factor results in a small ratio of flipping weights thus leading to little information gain in the training process [21, 37][2].

In this paper, we propose a Rotated Binary Neural Network (RBNN) to further mitigate the quantization error from the intrinsic angular bias as illustrated in Fig. 3. To the best of our knowledge, this is the first work that explores and reduces the influence of angular bias on quantization error in the field of BNN. To this end, we devise an angle alignment scheme by learning a rotation matrix that rotates the full-precision weight vector to its geometrical vertex of the binary hypercube at the beginning of each training epoch. Instead of directly learning a large rotation matrix, we introduce a bi-rotation formulation that learns two smaller matrices with a significantly reduced complexity. A series of optimization steps are then developed to learn the rotation matrices and binarization alternatingly to align the angle difference as shown in Fig. 2(a), which significantly reduces the quantization error as illustrated in Fig. 2(b). To get rid of the possible local optimum in the optimization, we dynamically adjust the rotated weights for binarization in the training stage. We show that the proposed rotation not only reduces the angular bias which leads to less quantization error, but also achieves around 50% weight flips thereby achieving maximum information gain. Finally, we provide a training-aware approximation of the sign function for gradient backpropagation. We show the superiority of RBNN through extensive experiments.

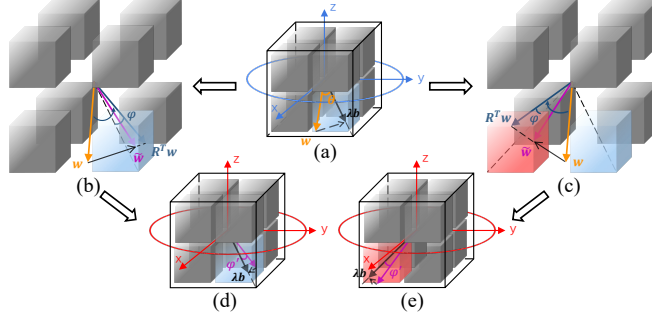

Figure 3: Framework of our RBNN. The weight vector is rotated at the beginning of each training epoch such that the angular bias $\varphi$ between the rotated weight vector and the geometrical binary vertex is smaller than that of the original $\theta$. After rotation, the weights are either unflipped (b) or flipped (c) which increases the information gain. During training, the rotated weights are dynamically adjusted such that $\tilde{\mathbf{w}}$ with much less angular bias $\varphi'$ is obtained, which then follows up the binarization.

## 2   Related Work

The pioneering BNN work dates back to [9] that binarizes the weights and activations to $-1$ or $+1$ by the sign function. The straight-through estimator (STE) [3] was proposed for the gradient backpropagation. Following this, abundant works have been devoted to improving the accuracy performance and implementing them in low-power and resource-constrained platforms. We refer readers to the survey paper [40, 36] for a more detailed overview.

XNOR-Net [38] includes all the basic components from [3] but further introduces a per-channel scaling factor to reduce the quantization error. The scaling factor is obtained through the $\ell_1$-norm of both weights and activations before binarization. DoReFa-Net [45] introduces a changeable bit-width for the quantization of weights and activations, even the gradient in the backpropagation. The scaling factor is layer-wise and deduced from network weights which allows efficient inference since the weights do not change after training. XNOR++ [5] fuses the separate activation and weight scaling factors in [3] into a single one which is then learned discriminatively via backpropagation.

Besides using the scaling factor to reduce the quantization error, more recent works are engaged in expanding the expressive ability to gain more information in the learning of BNN and devising a differentiable approximation to the sign function to achieve the propagation of the gradient. ABC-Net [31] proposes multiple parallel binary convolution layers to enhance the model accuracy. Bi-Real Net [32] adds ResNet-like shortcuts to reduce the information loss caused by binarization. Both ABC-Net and Bi-Real Net modify the structure of networks to strengthen the information of BNN learning. However, they ignore the probability of weight flips, thus the actual learning capacities of ABC-Net and Bi-Real Net are smaller than $2^n$ as stressed in Sec. 1. To compensate, [21, 37] strengthen the learning ability of BNNs during network training via increasing the probability of weight flips.

## 3   Approach

### 3.1   Binary Neural Networks

Given a CNN model, we denote $\mathbf{w}^i \in \mathbb{R}^{n^i}$ and $\mathbf{a}^i \in \mathbb{R}^{m^i}$ as its weights and feature maps in the $i$-th layer, where $n^i = c^i_{out} \cdot c^i_{in} \cdot w^i_f \cdot h^i_f$ and $m^i = c^i_{out} \cdot w^i_a \cdot h^i_a$. $(c^i_{out}, c^i_{in})$ represent the number of output and input channels, respectively. $(w^i_f, h^i_f)$ and $(w^i_a, h^i_a)$ are the width and height of filters and feature maps, respectively. We then have

$$\mathbf{a}^i = \mathbf{w}^i \circledast \mathbf{a}^{i-1},$$

where $\circledast$ is the standard convolution and we omit activation layers for simplicity. The BNN aims to convert $\mathbf{w}^i$ and $\mathbf{a}^i$ into $\mathbf{b}^i_w \in \{-1, +1\}^{n^i}$ and $\mathbf{b}^i_a \in \{-1, +1\}^{m^i}$, such that the convolution can be achieved by using the efficient xnor and bitcount logics. Following [23, 5], we binarize the activations

with the sign function:
$$\mathbf{b}_a^i = \text{sign}(\mathbf{b}_w^i \odot \mathbf{b}_a^{i-1}),$$
where $\odot$ represents the xnor and bitcount logics, and $\text{sign}(\cdot)$ denotes the sign function which returns 1 if the input is larger than zero, and $-1$ otherwise. Similar to [23, 45, 5], $\mathbf{b}_w^i$ can also be obtained by $\mathbf{b}_w^i = \text{sign}(\mathbf{w}^i)$ and a scaling factor can be applied to compensate for the norm difference.

However, the existence of angular bias between $\mathbf{w}^i$ and $\mathbf{b}_w^i$ could lead to large quantization error as analyzed in Sec. 1. Besides, it results in consistent signs between $\mathbf{w}^i$ and $\mathbf{b}_w^i$ which lessens the information gain (see footnote 1). We aim to minimize this angular bias to reduce the quantization error, meanwhile increasing the probability of weight flips to increase the information gain.

### 3.2  Rotated Binary Neural Networks

As shown in Fig. 3, we consider applying a rotation matrix $\mathbf{R}^i \in \mathbb{R}^{n^i \times n^i}$ to $\mathbf{w}^i$ at the beginning of each training epoch, such that the angle $\varphi^i$ between the rotated weight vector $(\mathbf{R}^i)^T \mathbf{w}^i$ and its binary vector $\text{sign}\big((\mathbf{R}^i)^T \mathbf{w}^i\big)$ should be minimized. To this end, we derive the following formulation:

$$\cos(\varphi^i) = \frac{\text{sign}\big((\mathbf{R}^i)^T \mathbf{w}^i\big)^T \big((\mathbf{R}^i)^T \mathbf{w}^i\big)}{\|\text{sign}\big((\mathbf{R}^i)^T \mathbf{w}^i\big)\|_2 \|(\mathbf{R}^i)^T \mathbf{w}^i\|_2}, \quad s.t. \quad (\mathbf{R}^i)^T \mathbf{R}^i = \mathbf{I}_{n^i}, \tag{3}$$

where $\mathbf{I}_{n^i} \in \mathbb{R}^{n^i \times n^i}$ is an $n^i$-th order identity matrix. It is easy to know that $\|\text{sign}\big((\mathbf{R}^i)^T \mathbf{w}^i\big)\|_2 = \sqrt{n^i}$, $\|(\mathbf{R}^i)^T \mathbf{w}^i\|_2 = \|\mathbf{w}^i\|_2$, both of which are constant[3]. Thus, Eq. (3) can be further simplified as

$$\cos(\varphi^i) = \eta^i \cdot \text{sign}\big((\mathbf{R}^i)^T \mathbf{w}^i\big)^T \big((\mathbf{R}^i)^T \mathbf{w}^i\big) = \eta^i \cdot \text{tr}\big(\mathbf{b}_{w'}^i (\mathbf{w}^i)^T \mathbf{R}^i\big), \ s.t. \ (\mathbf{R}^i)^T \mathbf{R}^i = \mathbf{I}_{n^i}, \tag{4}$$

where $\eta^i = 1/\big(\|\text{sign}\big((\mathbf{R}^i)^T \mathbf{w}^i\big)\|_2 \|(\mathbf{R}^i)^T \mathbf{w}^i\|_2\big) = 1/\big(\sqrt{n^i}\|\mathbf{w}^i\|_2\big)$, $\text{tr}(\cdot)$ returns the trace of the input matrix and $\mathbf{b}_{w'}^i = \text{sign}\big((\mathbf{R}^i)^T \mathbf{w}^i\big)$. However, Eq. (4) involves a large rotation matrix, $n^i$ of which can be up to millions in a neural network. Direct optimization of $\mathbf{R}^i$ would consume massive memory and computation. Besides, performing a large rotation leads to $\mathcal{O}\big((n^i)^2\big)$ complexity in both space and time.

To deal with this, inspired by the properties of Kronecker product [27], we introduce a bi-rotation scenario where two smaller rotation matrices $\mathbf{R}_1^i \in \mathbb{R}^{n_1^i \times n_1^i}$ and $\mathbf{R}_2^i \in \mathbb{R}^{n_2^i \times n_2^i}$ are used to reconstruct the large rotation matrix $\mathbf{R}^i \in \mathbb{R}^{n^i \times n^i}$ with $n^i = n_1^i \cdot n_2^i$. One of the basic property of Kronecker product [27] is that if two matrices $\mathbf{R}_1^i \in \mathbb{R}^{n_1^i \times n_1^i}$ and $\mathbf{R}_2^i \in \mathbb{R}^{n_2^i \times n_2^i}$ are orthogonal, then $\mathbf{R}_1^i \otimes \mathbf{R}_2^i \in \mathbb{R}^{n_1^i n_2^i \times n_1^i n_2^i}$ is orthogonal as well, where $\otimes$ denotes the Kronecker product. Another basic property of Kronecker product comes at:

$$(\mathbf{w}^i)^T (\mathbf{R}_1^i \otimes \mathbf{R}_2^i) = \text{Vec}\big((\mathbf{R}_2^i)^T (\mathbf{W}^i)^T \mathbf{R}_1^i\big), \tag{5}$$

where $\text{Vec}(\cdot)$ vectorizes its input and $\text{Vec}(\mathbf{W}^i) = \mathbf{w}^i$. Thus, we can see that applying the bi-rotation to $\mathbf{W}^i$ is equivalent to applying an $\mathbf{R}^i = \mathbf{R}_1^i \otimes \mathbf{R}_2^i \in \mathbb{R}^{n_1^i n_2^i \times n_1^i n_2^i}$ rotation to $\mathbf{w}^i$. Learning two smaller matrices $\mathbf{R}_1^i$ and $\mathbf{R}_2^i$ can well reconstruct the large rotation matrix $\mathbf{R}^i$. Moreover, performing the bi-rotation consumes only $\mathcal{O}\big((n_1^i)^2 + (n_2^i)^2\big)$ space complexity and $\mathcal{O}\big((n_1^i)^2 n_2^i + n_1^i (n_2^i)^2\big)$ time complexity, respectively, leading to a significant complexity reduction compared to the large rotation[4]. Accordingly, Eq. (4) can be reformulated as

$$\cos(\varphi^i) = \eta^i \cdot \text{tr}\Big(\mathbf{b}_{w'}^i \text{Vec}\big((\mathbf{R}_2^i)^T (\mathbf{W}^i)^T \mathbf{R}_1^i\big)\Big) = \eta^i \cdot \text{tr}\big(\mathbf{B}_{W'}^i (\mathbf{R}_2^i)^T (\mathbf{W}^i)^T \mathbf{R}_1^i\big),$$
$$s.t. \qquad (\mathbf{R}_1^i)^T \mathbf{R}_1^i = \mathbf{I}_{n_1^i}, \quad (\mathbf{R}_2^i)^T \mathbf{R}_2^i = \mathbf{I}_{n_2^i}, \tag{6}$$

where $\mathbf{B}_{W'}^i = \text{sign}\big((\mathbf{R}_1^i)^T \mathbf{W}^i \mathbf{R}_2^i\big)$. Finally, we rewrite our optimization objective below:

$$\underset{\mathbf{B}_{W'}^i, \mathbf{R}_1^i, \mathbf{R}_2^i}{\arg\max} \ \text{tr}\big(\mathbf{B}_{W'}^i (\mathbf{R}_2^i)^T (\mathbf{W}^i)^T \mathbf{R}_1^i\big),$$
$$s.t. \ \ \mathbf{B}_{W'}^i \in \{-1, +1\}^{n_1^i \times n_2^i}, \ (\mathbf{R}_1^i)^T \mathbf{R}_1^i = \mathbf{I}_{n_1^i}, \ (\mathbf{R}_2^i)^T \mathbf{R}_2^i = \mathbf{I}_{n_2^i}. \tag{7}$$

## 3.3 Alternating Optimization

Eq. (7) is non-convex *w.r.t.* $\mathbf{B}_{W'}^i$, $\mathbf{R}_1^i$ and $\mathbf{R}_2^i$. To find a feasible solution, we adopt an alternating optimization approach, *i.e.*, updating one variable with the rest two fixed until convergence.

**1) $\mathbf{B}_{W'}^i$-step:** Fix $\mathbf{R}_1^i$ and $\mathbf{R}_2^i$, then learn the binarization $\mathbf{B}_{W'}^i$. The sub-problem of Eq. (7) becomes:

$$\arg\max_{\mathbf{B}_{W'}^i} \operatorname{tr}\big(\mathbf{B}_{W'}^i(\mathbf{R}_2^i)^T(\mathbf{W}^i)^T\mathbf{R}_1^i\big), \quad s.t. \quad \mathbf{B}_{W'}^i \in \{-1,+1\}^{n_1^i \times n_2^i}, \tag{8}$$

which can be achieved by $\mathbf{B}_{W'}^i = \operatorname{sign}\big((\mathbf{R}_1^i)^T\mathbf{W}^i\mathbf{R}_2^i\big)$.

**2) $\mathbf{R}_1^i$-step:** Fix $\mathbf{B}_{W'}^i$ and $\mathbf{R}_2^i$, then update $\mathbf{R}_1^i$. The corresponding sub-problem is:

$$\arg\max_{\mathbf{R}_1^i} \operatorname{tr}\big(\mathbf{G}_1^i\mathbf{R}_1^i\big), \quad s.t. \quad (\mathbf{R}_1^i)^T\mathbf{R}_1^i = \mathbf{I}_{n_1^i}, \tag{9}$$

where $\mathbf{G}_1^i = \mathbf{B}_{W'}^i(\mathbf{R}_2^i)^T(\mathbf{W}^i)^T$. The above maximum can be achieved by using the polar decomposition [34]: $\mathbf{R}_1^i = \mathbf{V}_1^i(\mathbf{U}_1^i)^T$, where $\mathbf{G}_1^i = \mathbf{U}_1^i\mathbf{S}_1^i(\mathbf{V}_1^i)^T$ is the SVD of $\mathbf{G}_1^i$.

**3) $\mathbf{R}_2^i$-step:** Fix $\mathbf{B}_{W'}^i$ and $\mathbf{R}_1^i$, then update $\mathbf{R}_2^i$. The corresponding sub-problem becomes

$$\arg\max_{\mathbf{R}_2^i} \operatorname{tr}\big((\mathbf{R}_2^i)^T\mathbf{G}_2^i\big), \quad s.t. \quad (\mathbf{R}_2^i)^T\mathbf{R}_2^i = \mathbf{I}_{n_2^i}, \tag{10}$$

where $\mathbf{G}_2^i = (\mathbf{W}^i)^T\mathbf{R}_1^i\mathbf{B}_{W'}^i$. Similar to the updating rule for $\mathbf{R}_1^i$, the updating rule for $\mathbf{R}_2^i$ is $\mathbf{R}_2^i = \mathbf{U}_2^i(\mathbf{V}_2^i)^T$, where $\mathbf{G}_2^i = \mathbf{U}_2^i\mathbf{S}_2^i(\mathbf{V}_2^i)^T$ is the SVD of $\mathbf{G}_2^i$.

In the experiments, we iteratively update $\mathbf{B}_{W'}^i$, $\mathbf{R}_1^i$ and $\mathbf{R}_2^i$, which can reach convergence after three cycles of updating. Therefore, the weight rotation can be efficiently implemented.

## 3.4 Adjustable Rotated Weight Vector

We narrow the angular bias between the full-precision weights and the binarization using our bi-rotation at the beginning of each training epoch. Then, we can set $\tilde{\mathbf{w}}^i = (\mathbf{R}^i)^T\mathbf{w}^i$, which will be fed to the sign function and follow up the standard gradient update in the neural network. However, the alternating optimization may get trapped in a local

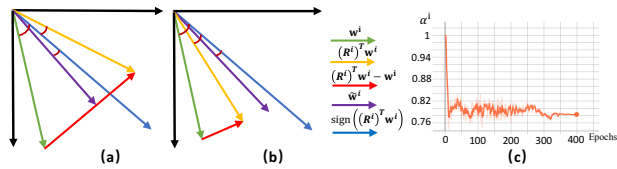

Figure 4: Best viewed with zooming in.

optimum that either overshoots (Fig. 4(a)) or undershoots (Fig. 4(b)) the binarization $\operatorname{sign}\big((\mathbf{R}^i)^T\mathbf{w}^i\big)$. To deal with this, we further propose to self-adjust the rotated weight vector as below:

$$\tilde{\mathbf{w}}^i = \mathbf{w}^i + \big((\mathbf{R}^i)^T\mathbf{w}^i - \mathbf{w}^i\big) \cdot \alpha^i, \tag{11}$$

where $\alpha^i = \operatorname{abs}\big(\sin(\beta^i)\big) \in [0,1]$ and $\beta^i \in \mathbb{R}$.

As can be seen from Fig. 4, Eq. (11) constrains that the final weight vector moves along the residual direction of $(\mathbf{R}^i)^T\mathbf{w}^i - \mathbf{w}^i$ with $\alpha^i \geq 0$. It is intuitive that when overshooting, $\alpha^i \leq 1$; when undershooting, $\alpha^i \geq 1$. We empirically observe that overshooting is in a dominant position. Thus, we simply constrain $\alpha^i \in [0,1]$ to shrink the feasible region of $\alpha^i$, which we find can well further reduce the quantization error and boost the performance as demonstrated in Table 4. The final value of $\alpha^i$ varies across different layers. In Fig. 4(c), we show a toy example of how $\alpha^i$ updates during training in ResNet-20 (layer2.2.conv2).

At the beginning of each training epoch, with fixed $\mathbf{w}^i$, we learn the rotation matrix $\mathbf{R}^i$ ($\mathbf{R}_1^i$ and $\mathbf{R}_2^i$ actually). In the training stage, with fixed $\mathbf{R}^i$, we feed the sign function using Eq. (11) in the forward, and update $\mathbf{w}^i$ and $\beta^i$ in the backward.

## 3.5 Gradient Approximation

The derivative of the sign function is almost zero everywhere, which makes the training unstable and degrades the accuracy performance. To solve it, various gradient approximations in the literature

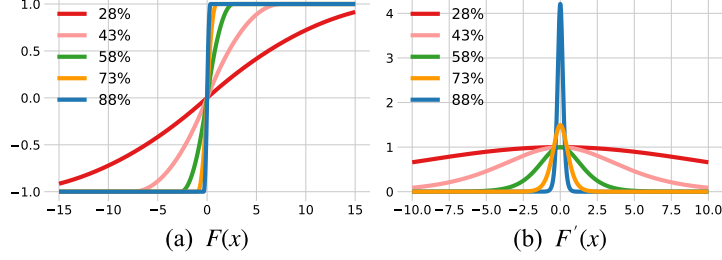

<div align="center">(a) $F(x)$        (b) $F'(x)$</div>

Figure 5: Visualization of our approximation function and its derivative *w.r.t.* different values of $\frac{e}{E}$.

have been proposed to enable the gradient updating, *e.g.*, straight through estimation [3], piecewise polynomial function [32], annealing hyperbolic tangent function [1], error decay estimator [37] and so on. Instead of simply using these existing approximations, in this paper, we further devise the following training-aware approximation function to replace the sign function:

$$F(x) = \begin{cases} k \cdot \left( - \operatorname{sign}(x)\frac{t^2 x^2}{2} + \sqrt{2}tx \right), & |x| < \frac{\sqrt{2}}{t}, \\ \operatorname{sign}(x) \cdot k, & \text{else.} \end{cases} \quad (12)$$

with

$$t = 10^{T_{\min} + \frac{e}{E} \cdot (T_{\max} - T_{\min})}, \quad k = \max(\frac{1}{t}, 1),$$

where $T_{\min} = -2$, $T_{\max} = 1$ in our implementation, $E$ is the number of training epochs and $e$ represents the current epoch. As can be seen, the shape of $F(x)$ relies on the value of $\frac{e}{E}$, which indicates the training progress. Then, the gradient of $F(x)$ *w.r.t.* the input $x$ can be obtained by

$$F'(x) = \frac{\partial F(x)}{\partial x} = \max \left( k \cdot (\sqrt{2}t - |t^2 x|), 0 \right). \quad (13)$$

In Fig. 5, we visualize Eq. (12) and Eq. (13) *w.r.t.* varying values of $\frac{e}{E}$. In the early training stage, the gradient exists almost everywhere, which overcomes the drawback of sign function and enables the updating of the whole network. As the training proceeds, our approximation gradually becomes the sign-like function, which ensures the binary property. Thus, our approximation is training-aware. So far, we can have the gradient of loss function $\mathcal{L}$ *w.r.t.* the activation $\mathbf{a}^i$ and weight $\mathbf{w}^i$:

$$\sigma_{\mathbf{a}^i} = \frac{\partial \mathcal{L}}{\partial F(\mathbf{a}^i)} \cdot \frac{\partial F(\mathbf{a}^i)}{\partial \mathbf{a}^i}, \quad \sigma_{\mathbf{w}^i} = \frac{\partial \mathcal{L}}{\partial F(\tilde{\mathbf{w}}^i)} \cdot \frac{\partial F(\tilde{\mathbf{w}}^i)}{\partial \tilde{\mathbf{w}}^i} \cdot \frac{\partial \tilde{\mathbf{w}}^i}{\partial \mathbf{w}^i}, \quad (14)$$

where

$$\frac{\partial \tilde{\mathbf{w}}^i}{\partial \mathbf{w}^i} = (1 - \alpha^i) \cdot \mathbf{I}_{n^i} + \alpha^i \cdot (\mathbf{R}^i)^T. \quad (15)$$

Besides, the gradient of $\alpha^i$ in Eq. (11) can be obtained by

$$\sigma_{\alpha^i} = \frac{\partial \mathcal{L}}{\partial \alpha^i} = \frac{\partial \mathcal{L}}{\partial \tilde{\mathbf{w}}^i} \frac{\partial \tilde{\mathbf{w}}^i}{\partial \alpha^i} = \sum_j \left[ \sigma_{\tilde{\mathbf{w}}^i} \left( (\mathbf{R}^i)^T \mathbf{w}^i - \mathbf{w}^i \right) \right]_j, \quad (16)$$

where $[\cdot]_j$ denotes the $j$-th element of its input vector. Note that the error decay estimator in [37] can be also regarded as a training-aware approximation. However, our design in Eq. (12) is fundamentally different from that in [37] and its superiority is validated in Sec. 4.2.

## 4 Experiments

In this section, we evaluate our RBNN on CIFAR-10 [25] using ResNet-18/20 [19] and VGG-small [44], and on ImageNet [11] using ResNet-18/34 [19]. Following the compared methods, all convolutional and fully-connected layers except the first and last ones are binarized. We implement RBNN with Pytorch and the SGD is adopted as the optimizer. Also, for fair comparison, we only apply the classification loss during training.

Table 1: Performance comparison with SOTAs on CIFAR-10. W/A denotes the bit length of weights and activations. The $\star$ denotes the network with the Bi-Real structure [32].

| Network | Method | W/A | Acc |
|---|---|---|---|
| ResNet-18 | FP | 32/32 | 93.0% |
| | RAD [13] | 1/1 | 90.5% |
| | IR-Net [37] | 1/1 | 91.5% |
| | **RBNN(Ours)** | 1/1 | **92.2%** |
| ResNet-20 | FP | 32/32 | 91.7% |
| | DoReFa [45] | 1/1 | 79.3% |
| | DSQ [15] | 1/1 | 84.1% |
| | IR-Net [37] | 1/1 | 85.4% |
| | **RBNN(Ours)** | 1/1 | **86.5%** |
| | IR-Net$^\star$ [37] | 1/1 | 86.5% |
| | **RBNN$^\star$(Ours)** | 1/1 | **87.8%** |
| VGG-small | FP | 32/32 | 91.7% |
| | LAB [22] | 1/1 | 87.7% |
| | XNOR-Net [38] | 1/1 | 89.8% |
| | BNN [23] | 1/1 | 89.9% |
| | RAD [13] | 1/1 | 90.0% |
| | IR-Net [37] | 1/1 | 90.4% |
| | **RBNN(Ours)** | 1/1 | **91.3%** |

Table 2: Performance comparison with SOTAs on ImageNet. W/A denotes the bit length of weights and activations. We report the top-1 and top-5 accuracy performances.

| Network | Method | W/A | Top-1 | Top-5 |
|---|---|---|---|---|
| ResNet-18 | FP | 32/32 | 69.6% | 89.2% |
| | ABC-Net [31] | 1/1 | 42.7% | 67.6% |
| | XNOR-Net [38] | 1/1 | 51.2% | 73.2% |
| | BNN+ [10] | 1/1 | 53.0% | 72.6% |
| | DoReFa [45] | 1/2 | 53.4% | - |
| | Bi-Real [32] | 1/1 | 56.4% | 79.5% |
| | XNOR++ [5] | 1/1 | 57.1% | 79.9% |
| | IR-Net [37] | 1/1 | 58.1% | 80.0% |
| | **RBNN(Ours)** | 1/1 | **59.9%** | **81.9%** |
| ResNet-34 | FP | 32/32 | 73.3% | 91.3% |
| | ABC-Net [31] | 1/1 | 52.4% | 76.5% |
| | Bi-Real [32] | 1/1 | 62.2% | 83.9% |
| | IR-Net [37] | 1/1 | 62.9% | 84.1% |
| | **RBNN(Ours)** | 1/1 | **63.1%** | **84.4%** |

## 4.1 Experimental Results

### 4.1.1 CIFAR-10

On CIFAR-10, we compare our RBNN with several SOTAs. For ResNet-18, we compare with RAD [13] and IR-Net [37]. For ResNet-34, we compare with DoReFa [45], DSQ [15], and IR-Net [37]. For VGG-small, we compare with LAB [22], XNOR-Net [38], BNN [23], RAD [13], and IR-Net [37]. We list the experimental results in Table 1. As can be seen, RBNN consistently outperforms the SOTAs. Compared with the best baseline [37], RBNN achieves 0.7%, 1.1% and 0.9% accuracy improvements with ResNet-18, ResNet-20 with normal structure [19], and VGG-small, respectively. Furthermore, binarizing network with the Bi-Real structure [32] achieves a better accuracy performance over the normal structure as shown by ResNet-20. For example, with the Bi-Real structure, IR-Net obtains 1.1% accuracy improvements while RBNN also gains 1.3% improvements. Other variants of network structure proposed in [4, 46, 16] and training loss in [22, 13, 41, 17] can be combined to further improve the final accuracy performance. Nevertheless, under the same structure, our RBNN performs the best (87.8% of RBNN *v.s.* 86.5% of IR-Net for ResNet-20 with Bi-Real structure). Hence, the superiority of the angle alignment is evident.

### 4.1.2 ImageNet

We further show the experimental results on ImageNet in Table 2. For ResNet-18, we compare RBNN with ABC-Net [31], XNOR-Net [38], BNN+ [10], DoReFa [45], Bi-Real [32], XNOR++ [5], and IR-Net [37]. For ResNet-34, ABC-Net [31], Bi-Real [32], and IR-Net [37] are compared. As shown in Table 2, RBNN beats all the compared binary models in both top-1 and top-5 accuracy. More detailedly, with ResNet-18, RBNN achieves 59.9% and 81.9% in top-1 and top-5 accuracy, with 1.8% and 1.9% improvements over IR-Net, respectively. With ResNet-34, it achieves a top-1 accuracy of 63.1% and a top-5 accuracy of 84.4%, with 0.2% and 0.3% improvements over IR-Net, respectively.

## 4.2 Performance Study

In this section, we first show the benefit of our training-aware approximation over other recent advances [3, 32, 37]. And then, we show the effect of different components proposed in our RBNN. All the experiments are conducted on top of ResNet-20 with Bi-Real structure on CIFAR-10.

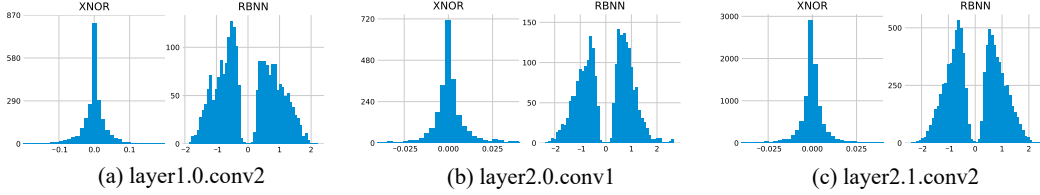

(a) layer1.0.conv2        (b) layer2.0.conv1        (c) layer2.1.conv2

Figure 6: Weight histograms (before binarization) of the XNOR and RBNN in ResNet-20.

Table 3: Gradient Approximation Analysis.

| Method | W/A | Acc |
|--------|-----|-----|
| FP | 32/32 | 91.7% |
| STE [3] | 1/1 | 84.9% |
| PPF [32] | 1/1 | 86.9% |
| EDE [37] | 1/1 | 86.0% |
| Ours | 1/1 | 87.8% |

Table 4: Ablation Study of RBNN. B, T, R and A respectively denote binarization using XNOR-Net, training-aware approximation, weight rotation and adjustable scheme.

| Method | W/A | Acc |
|--------|-----|-----|
| FP | 32/32 | 91.7% |
| B | 1/1 | 83.7% |
| B + R | 1/1 | 86.4% |
| B + T | 1/1 | 86.6% |
| B + T + R | 1/1 | 87.1% |
| B + T + R + A (RBNN) | 1/1 | 87.8% |

Table 3 compares the performances of RBNN based on various gradient approximations including straight through estimation (denoted by STE) [3], piecewise polynomial function (denoted by PPF) [32] and error decay estimator (denoted by EDE) [37]. As can be seen, STE shows the least accuracy. Though EDE also studies approximation with dynamic changes, its performance is even worse than PPF which is a fixed approximation. In contrast, our training-aware approximation achieves 1.8% improvements over EDE, which validates the effectiveness of our approximation.

To further understand the effect of each component in our RBNN, we conduct an ablation study by starting with the binarization using XNOR-Net [38] (denoted by B), and then gradually add different parts of training-aware approximation (denoted by T), weight rotation (denoted by R) and adjustable scheme (denoted by A). As shown in Table 4, the binarization using XNOR-Net suffers a great performance degradation of 8.0% compared with the full-precision model. By adding our weight rotation or training-aware approximation, the accuracy performance increases to 86.4% or 86.6%. Then, the collective effort of weight rotation and training-aware approximation further raises it to 87.1%. Lastly, by considering the adjustable weight vector in the training process, our RBNN achieves the highest accuracy of 87.8%. Therefore, each part of RBNN plays its unique role in improving the performance.

## 4.3 Weight Distribution and Flips

Fig. 6 shows the histograms of weights (before binarization) for XNOR-Net and our RBNN. It can be seen that the weight values for XNOR-Net are mixed up tightly around zero center and the value magnitude remains far less than 1. Thus it causes large quantization error when being pushed to the binary values of −1 and +1. On the contrary, our RBNN results in two-mode distributions, each of which is centered around −1/ + 1. Besides, there exist few weights around the zero, which creates a clear boundary between the two distributions. Thus, by the weight rotation, our RBNN effectively reduces quantization error as explained in Fig. 2.

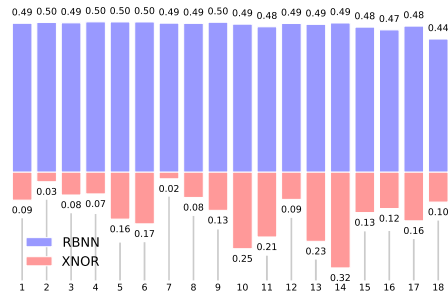

Figure 7: Weight flip rates of our RBNN and XNOR-Net in different layers of ResNet-20.

As discussed in Sec. 2, the capacity of learning BNN is up to $2^n$ where $n$ is the total number of weight elements. When the probability of each element being $-1$ or $+1$ is equal during training, it reaches the maximum of $2^n$. We compare the initialization weights and binary weights, and then show the weight flipping rates of our RBNN and XNOR-Net across the layers of ResNet-20 in Fig. 7. As can be observed, XNOR-Net leads to a small flipping rate, *i.e.*, most positive weights are directly quantized to $+1$, and vice versa. Differently, RBNN leads to around 50% weight flips each layer due to the introduced weight rotation as illustrated in Fig. 3, which thus maximizes the information gain during the training stage.

## 5 Conclusion

In this paper, we analyzed the influence of angular bias on the quantization error in binary neural networks and proposed a Rotated Binary Neural Network (RBNN) to achieve the angle alignment between the rotated weight vector and the binary vector at the beginning of each training epochs. We have also introduced a bi-rotation scheme involving two smaller rotation matrices to reduce the complexity of learning a large rotation matrix. In the training stage, our method dynamically adjusts the rotated weight vector via backward gradient updating to overcome the potential sub-optimal problem in the optimization of bi-rotation. Our rotation maximizes the information gain of learning BNN by achieving around 50% weight flips. To enable gradient propagation, we have devised a training-aware approximation of the sign function. Extensive experiments have demonstrated the efficacy of our RBNN in reducing the quantization error, and its superiorities over several SOTAs.

## Acknowledgement

This work is supported by the National Natural Science Foundation of China (No.U1705262, No.61772443, No.61572410, No.61802324 and No.61702136), National Key R&D Program (No.2017YFC0113000, and No.2016YFB1001503), Key R&D Program of Jiangxi Province (No. 20171ACH80022), Natural Science Foundation of Guangdong Province in China (No.2019B1515120049) and National Key R&D Plan Project (No.2018YFC0830105 and No.2018YFC0830100).

## Broader Impact

**Benefit:** The binary neural network community may benefit from our research. The proposed Rotated Binary Neural Network (RBNN) provides a novel perspective to lessen the quantization error by reducing the angular bias, which was ignored by previous works. With the code publicly available, our work will also help researchers quantize DNNs so that the deep models can be deployed on devices with limited resources such as mobile phones.

**Disadvantage:** The angular bias between the activation and its binarization remains an open problem. It may be not appropriate to apply our rotation to the activation vector since it will add the computation in the inference.

**Consequence:** The failure of the network quantization will not bring serious consequences, as our RBNN causes fewer accuracy drops compared to other SOTAs.

**Data Biases:** The proposed RBNN is irrelevant to data selection, so it does not have the data bias problem.

## Footnotes

*Corresponding Author: rrji@xmu.edu.cn

[2]The binarization in Eq. (1) is obtained by $\mathbf{b} = \text{sign}(\mathbf{w})$, which does not change the coordinate quadrant as shown in Fig. 1. Thus, only a small number of weight flips occur in the training stage. See Sec. 4.3 for our experimental validation.

[3]To stress, the rotation is applied at the beginning of each training epoch instead of the training stage. Thus, $\|\mathbf{w}^i\|_2$ should be regarded as a constant.

[4]With $n_1^i \cdot n_2^i = n^i$, our RBNN achieves the least complexity when $n_1^i = n_2^i = \sqrt{n^i}$, which is also our experimental setting.

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
