[Reviews · NeurIPS 2020]

Review 1

Summary and Contributions: This paper focuses on solving the quantization error in binary neural network. Their contributions are: 1、This paper for the first time talks about the quantization error from the angular bias between the weight vector and its binarization. 2、To solve the angular bias, this paper introduces a weight rotation scheme in the beginning of each training epoch. A bi-rotation is further devised to reduce the optimization complexity. The optimization objective is intuitive and the derivations are elaborated. Experiments have verified the effectiveness of closing angular bias. 3、 A training-aware approximation is proposed to compensate the “no gradient” problem in the sign function. Its superiority over existing methods is experimentally validated. 4、 Extensive experiments on CIFAR-10 and ImageNet show better performance of the proposed method.

Strengths: Soundness of the claims: To my knowledge, this is the first work that talks about the quantization error from angular bias in BNN. I think the angular bias indeed exists from a mathematical view and it has been validated by the authors. The discovery is insightful. Their solution of weight rotation is very interesting and attractive. I have carefully checked all the optimization formulations and confirmed their correctness. The training-aware approximation may not be a pioneer but shows significant difference and superiorities over the old methods. Experiments have been made to show its better performance. Significance and novelty of the contribution: In general, I agree with the novelty and contributions claimed in this paper. This paper is well written and easy to follow. The discussion on angular bias is the first time. Their solution of weight rotation is original and attracts my attention. Experimental results provide comparable accuracy performance and the available code is a bonus to the community. Relevance to the NeurIPS community: It is highly relevant. I have seen many related works accepted by NeurIPS each year.

Weaknesses: I still need some clarifications from the authors. The concerns are listed below. 1、 It is unclear why ${\alpha}^i$ in Eq. (11) is limited within [0, 1]. In my opinion, the final weight vector is not necessary to be among the original weight and rotated one. Please clarify. 2、 The author performs rotations in the beginning of each training epoch. What if the rotation is applied each training iteration? Intuitively, this may feed back a better performance. 3、 There seems a contradiction between figure 2(b) and figure 5. In figure 2(b), the quantization error of the proposed method is almost zero for most layers. Obviously, figure 5 provides larger quantization error though the two-mode distributions of weight values are centered around -1/+1. 4、 As an analogy, the activation quantization also suffers from the angular bias. However, the authors simply binarize the activations using sign function (see line 99 – line 100). This seems counter-intuitive. 5、 Typo and grammar errors: Figure 2: and its binarizaton – and its binarization Line 59: with significantly reduced complexity – with a significantly reduced complexity

Correctness: Yes.

Clarity: Yes. A pleasure to read though some unclear parts in the weakness box need clarifying.

Relation to Prior Work: Yes.

Reproducibility: Yes

Additional Feedback: See the box for weakness. Post-rebuttal comments: The author has addressed my questions.


Review 2

Summary and Contributions: The paper proposes to learn rotation matrices to rotate the floating-point vectors in each layer before projecting them to binary vectors while learning binary neural networks. This approach reduces the angular bias in the float to binary projection. The results on cifar and imagenet show improved results compared to the previous methods when both weights and activations are binarized.

Strengths: 1. The idea of learning rotation matrices to reduce the angular bias when projecting from float to binary is new and interesting. 2. The paper is clearly written and the method is sufficiently explained. 3. Extensive experiments show improvements over the compared baselines.

Weaknesses: Some weaknesses/comments/questions of the proposed method in my opinion are as follows: 1) The improvement due to rotation seems small (ref. table 4) and the most of the improvement seems to be due to the gradient approximation. Please clarify. 2) How many additional learnable parameters are introduced while learning rotation matrices? Are they binary or floating point? Please provide a summary of new learnable parameters and actual (theoretical) improvement in memory and FLOPS. This is important to see as there are some parameters in the network is kept in floating point and additional parameters are introduced compared to the baselines. 3) What is the motivation for the specific form of gradient approximation (sec. 3.4) used in this paper? Why not simply use an existing function such as tanh and anneal using a temperature parameter to approximate the step function similar to [a]? 4) Is there any reason for explicitly learning the rotation matrices rather than penalizing the angle between floating-point and the binary vectors while learning? I believe, this approach will not have any additional learnable parameters and encourage the float vectors to align with the binary vectors. Please comment. 5) Is there any loss due to the introduction of the bi-rotation formulation? 6) What is the justification for Eq. 11? 7) Any overhead at test time? [a] Ajanthan, T., Gupta, K., Torr, P.H., Hartley, R. and Dokania, P.K., 2019. Mirror descent view for neural network quantization. arXiv preprint arXiv:1910.08237. Post rebuttal update: The rebuttal clarifies most of my initial concerns and authors have promised to add discussions in certain parts. The main one, improvement due to rotation on xnor-net (without gradient approximation) is interesting and I recommend including that in the paper. I increase the rating.

Correctness: The method and the claims are correct and the experimental setup is valid.

Clarity: Overall the paper is well written and sufficient details are provided. In addtion, please consider giving more details on the number of additional learnable parameters. See weakness.

Relation to Prior Work: Prior works are sufficiently discussed.

Reproducibility: Yes

Additional Feedback: Please release the code to ensure reproducibility.


Review 3

Summary and Contributions: The authors focus propose a method for training binary neural networks. They focus on an overlooked issue in the binarization process, namely the rotation bias, which degrades performance and propose a solution to it. They also propose a smooth gradient approximation function

Strengths: The paper is innovative in that it identifies something that no one has looked into before. The methodology is interesting. The insight is hard to realize, but the authors propose several interesting technical tricks to solve it. There is a good awareness of the literature.

Weaknesses: Context is sorely missing in the early parts of the paper. Intuition of what the paper is focusing on is very low in the introduction.

Correctness: It seems correct to me, although a more intuitive explanation in the introduction would have helped. The validation is correct. A doubt I am left with is what happens with the rotation - you cannot just rotate the weights without using the inverse rotation elsewhere, otherwise the network will give totally wrong predictions. So how do you compensate for rotating the weights? does it have any implication in terms of computational cost? because if you have to rotate the input tensor at every step to compensate for the weight rotation, then it is non-trivial. Please clarify in the rebuttal this point - or what I am misunderstanding. Two small things: - line 37: the scaling factor is on a per-channel basis, which is not clear in the text - line 86: Bi-real net does not increase the weights - line 104: "Besides, this..." I do not understand what the authors mean, even after reading the full text

Clarity: The paper has some issues early on, where the context and an intuitive explanation of what the authors are talking about would have helped a lot. Once they focus on the methodological part it is easier to follow. For example: - what the angular bias is is not explained at all. There is a graphical abstract (Fig 1.a) with very little information on it that I was unable to decode. Only after reading the rest of the paper I understood. - Th "weight flips" is equally impossible to understand early on. l50: "possibility of flips achieves 50%". First, I would say it is "probability", not possibility. But what changes for weights to flip? are we talking during iterations of training? across different images? it seems to me that what actually flips is the activations, not the weights themselves? anyway, I'm just hypothesizing, in any case this really needs a clarification early on. - The self-adjustment in eq. 11 has to be justified. Where does this formula come from? why does it work?

Relation to Prior Work: In general, the work shows good awareness of the literature and often discusses relevant work. Nit: Section 3.4, would be nice to include an explanation that several approximations do exist in the literature.

Reproducibility: Yes

Additional Feedback: Some suggestions going forward Please make a clear introduction of the concepts you're going to talk about. The paper would be so much more readable with a clear explanation of what's angular bias and where it comes from, and about the "flipping of weights". The graphical abstract should be way more intuitive too - a 2D sketch for example could do? I would have liked to see some formatting of the validation like (I know the content is there, but parsing it is harder): XNor-Net XNor-Net + Ours Bi-RealNet Bi-RealNet + Ours ... I think having a well thought-out optimization is particularly important to properly validate this method. It is mentioned by the authors that the angular bias could be corrected during optimization, but in practice it is shown that this is not the case. In my opinion, there has been a lot of improvement regarding optimization. Bi-Real net is a very good step forward, and it is great that it is included as a comparison. Yet, the current best baseline available is from "Training binary neural networks with real-to-binary convolutions", Martinez et al., ICLR'20 (what they call "strong baseline"), which has a large improvement over bi-real net just through refined optimization. In particular, I think that the two-stage training (proposed in another paper) is pivotal for a correct optimization. The question is, if you train the network "properly", will it correct the angular bias or still not manage? Is it possible to train the full rotation matrix? it might be impractical, but if done at least once we would know how close the "practical approach" is to it. Some of the notation used in tables (e.g. Table 4) is not explained in the caption. This means a reader has to go back and forth from table to text, which is annoying. It is always a nice touch to have "self-contained" tables if possible. the Broader Impact is kind of an extended "conclusions" section. Maybe not the aim of the neurips organizers? Rebuttal: The rebuttal didn't add much from my perspective. Most of my remarks were not aimed to be tackled for the revision - the authors are indeed correct, unfortunately time and resources are limited. Also rewriting parts of the text is not possible for a rebuttal. Thus I simply leave the marks as they were. I would however encourage the authors to put some extra effort polishing the aspects I mentioned - especially making an effort to give better context and intuition early on - as it will make the paper more attractive and approachable.


Review 4

Summary and Contributions: This paper explores the influence of angular bias on the quantization error. A Rotated Binary Neural Network (RBNN) is introduced to reduce the angular bias. Compared with previous work, the proposed method can further reduce the quantization error theoretically considering the angular bias. Experimental results on CIFAR-10 and ImageNet using different architectures demonstrate the effectiveness.

Strengths: 1. The main idea is novel and reasonable. The previous work adopt scaling factors to lessen the quantization error. I agree that scaling factor can only partly mitigates the problem and it is a right way to solve this problem by reducing the angular bias. 2. The bi-rotation formulation is interesting. It is a clever approach to leverage the property of Kronecker product to reduce computation complexity. I appreciate this technique. 3. Experimental results on CIFAR-10 dataset and ImageNet are promising. The proposed method can achieve state-of-the-art performance on various architectures. The ablation studies verify the effectiveness of different modules.

Weaknesses: 1. My main concern is about self-adjust parameter \alpha. In my opinion, the ideal value for \alpha is 1, i.e. \tilde{w} is equal to the rotated weights. However, during training, \alpha varies from 0 to 1 and I can not understand its physical meaning. For example, when \alpha equals to 0.5, does the quantization error of \tilde{w} is smaller than that of original weight w ? I think authors should give some proof. Also, the figure of how \alpha update during training is not given. Does \alpha equal to 1 at the last epoch ? 2. There is a typo: the term “G” in table 4 is not consistent with the term “T” in line 217.

Correctness: The proposed method is correct and reasonable.

Clarity: The paper is well writen and easy to understand.

Relation to Prior Work: The author claim that they are the first to explore the influence of angular bias on quantization error in the field of BNN, which is the main difference from previous works.

Reproducibility: Yes

Additional Feedback:

[Author Response · NeurIPS 2020]

**To All Reviewers:** With positive scores from all the reviewers, we thank all the reviewers for their valuable feedbacks.
Code has been submitted as the supplementary material and will be publicly released if the paper gets accepted.

**Q: Justification for self-adjustment of Eq. (11).** We design Eq. (11), $\tilde{\mathbf{w}}^i =$
$\mathbf{w}^i + \left((\mathbf{R}^i)^T \mathbf{w}^i - \mathbf{w}^i\right) \cdot \alpha^i$ to further mitigate angular bias. In Sec. 3.3, we adopt
alternating optimization for the non-convex objective of Eq. (7). It does reduce
the angular bias, which, however, cannot guarantee the global optimum. Usu-
ally, it either overshoots (Fig. A(a)) or undershoots (Fig. A(b)) the binarization
$\mathrm{sign}\left((\mathbf{R}^i)^T \mathbf{w}^i\right)$. Eq. (11) constrains that the final weight vector moves along the

Figure A: Viewed with zooming in.

residual direction of $(\mathbf{R}^i)^T \mathbf{w}^i - \mathbf{w}^i$ with $\alpha \geq 0$. It is intuitive that when overshooting, $\alpha \leq 1$; when undershooting,
$\alpha \geq 1$. We empirically observe that overshooting is in a dominant position. Thus, we simply constrain $\alpha \in [0, 1]$ to
shrink the learnable domain of $\alpha$, which we find can well further reduce the quantization error and boost the performance
as demonstrated in Tab. 4 of the paper. The final value of $\alpha$ varies across different layers. In Fig. A(c), we show how $\alpha$
updates during training in ResNet-20 (layer2.2.conv2). We will clarify this part in our final version.

**Reviewer #1:** clear accept (8) – insightful, interesting, attractive and pleasure to read.

**Q1: Per-iteration rotation.** Taking ResNet-20 on CIFAR-10 (one GTX 1080 GPU) as an example, per-iteration
rotation achieves 86.7% (53.33 hours) top-1 accuracy while per-epoch rotation achieves 86.5% (5.83 hours). Per-epoch
rotation achieves $9.15\times$ speed-up with negligible accuracy compromise. Thus we adopt per-epoch rotation. **Q2:**
**Fig. 2(b) and Fig. 5.** Fig. 2(b) shows per-layer quantization error consisting of $\lambda$, $\mathbf{w}$ and $\mathbf{b}$ in Eq. (1), and Fig. 5 shows
per-layer distribution of $\mathbf{w}$. Since $\lambda$ was ignored in Fig. 5, it is inappropriate to compute quantization error by comparing
$\mathbf{w}$ with their centers (binarization). **Q3: Angular bias of activation.** It consumes additional computation in the test
stage since the activation is input-related. Thus, we do not apply activation rotation. **Q4: Typos.** They have been fixed.

**Reviewer #2:** above the acceptance (6) – new, interesting, clearly written and sufficiently explained.

**Q1: Approximation brings more improvements.** Our approximation achieves a good accuracy of 86.6% based
on a mediocre baseline XNOR-Net (83.7%), which is hard to be further boosted. Nevertheless, our rotation still
increases it to 87.1%. Actually, combining our rotation with XNOR-Net (without approximation) can also obtain a
great performance of 86.4%. Thus, our rotation should not be evaluated simply by a relative increment from 86.6%
to 87.1%. **Q2: Additional parameters.** Additional parameters include $\alpha^i$, $\mathbf{R}_1^i$ and $\mathbf{R}_2^i$, which are floating-point. $\alpha^i$
brings negligible consumption. Theoretical complexities of $\mathbf{R}_1^i$ and $\mathbf{R}_2^i$ are provided in line 127 of the paper. We
take ResNet-20 on CIFAR-10 to test actual complexity. It takes 21 (19) seconds to finish one training epoch on one
GTX 1080 GPU with (without) bi-rotation. Besides, only 0.06MB space are introduced additionally. Both the time
and space cost are negligible. Moreover, $\mathbf{R}_1^i$ and $\mathbf{R}_2^i$ are applied in the training stage to learn weight binarization,
which means no overhead in the test stage. **Q3: Approximation motivation.** The approximation has to be "soft" to
backward gradient, and also has to well approximate the sign function to minimize the forward error. Though $\tanh$
function enables gradient propagation, it never reaches the exact values of -1/+1 even with annealing hyperparameter.
Our approach not only enables gradient propagation, but also behaves exactly the same as the sign function in the ends
which distinguishes our method from others. In our final version, we will reorganize Sec. 3.4 and include more related
works including $\tanh$ (annealing). **Q4: Penalizing the angle while learning.** Though reducing the angle error, we find
it hard to increase the probability of weight flip. Thus, the rotation is introduced. **Q5: loss of bi-rotation.** Bi-rotation
doesn't increase the loss since the bi-rotation matrices are equivalent to full-matrix rotation (see line 125 of the paper
and our response to **Q2** of Reviewer #3). **Q6: Test time overhead?** No (see **Q2**).

**Reviewer #3:** above the acceptance (6) – innovative, interesting, and good awareness of the literature.

**Q1: How to compensate for rotating the weights?** At the beginning of each training epoch, we apply the weight
rotation to reduce the angle first. A regular training epoch is then applied on basis of the rotated weights to retain
the accuracy (see lines 145–147 of the paper). **Q2: Going-forward suggestions.** We really appreciate your valuable
suggestions. First, since Martinez *et al.*, haven't released their code yet, we could not verify currently if properly
training the network will correct the angular bias as the rebuttal period is very limited. However, we will do it after the
rebuttal. Second, our GPU platform can perform full rotation on CIFAR-10 and it shows similar results to bi-rotation.
However, it exceeds our hardware capacity to evaluate on ImageNet because of the massive memory consumption.
In sum, our final version will include the following changes: 1) Angular bias and weight flip will be clearly defined
in the introduction. 2) The validation will be re-formatted and table notations will be explained in the caption. 3)
More existing approximations will be added in Sec. 3.4. 4) Lines 37, 86 and 104 will be rephrased. 5) Eq. (11) will be
justified. 6) We will rewrite our broader impact.

**Reviewer #4:** above the acceptance (6) – novel, interesting, well written and easy to understand.

**Q1: Typos.** We have carefully proofread the manuscript and fixed the typos.

[Meta-Review · NeurIPS 2020]

All four viewers provide favorable or very favorable reviews. The reviewers point out the novel ideas on solving the angular bias in binary neural networks, and point out the positive empirical results. The paper is therefore accepted.